# CONFORMATION-GUIDED MOLECULAR REPRESENTATION WITH HAMILTONIAN NEURAL NETWORKS

**Ziyao Li[1], Shuwen Yang[2]\*, Guojie Song[2]†, Lingsheng Cai[2]**
[1]Center for Data Science, Peking University, Beijing, China
[2]Key Laboratory of Machine Perception and Intelligence (MOE), Peking University, Beijing, China
{leeeezy,swyang,gjsong,cailingsheng}@pku.edu.cn

## ABSTRACT

Well-designed molecular representations (fingerprints) are vital to combine medical chemistry and deep learning. Whereas incorporating 3D geometry of molecules (i.e. conformations) in their representations seems beneficial, current 3D algorithms are still in infancy. In this paper, we propose a novel molecular representation algorithm which preserves 3D conformations of molecules with a Molecular Hamiltonian Network (HamNet). In HamNet, implicit positions and momentums of atoms in a molecule interact in the Hamiltonian Engine following the discretized Hamiltonian equations. These implicit coordinations are supervised with real conformations with translation- & rotation-invariant losses, and further used as inputs to the Fingerprint Generator, a message-passing neural network. Experiments show that the Hamiltonian Engine can well preserve molecular conformations, and that the fingerprints generated by HamNet achieve state-of-the-art performances on MoleculeNet, a standard molecular machine learning benchmark.

## 1 INTRODUCTION

The past several years have seen a prevalence of the intersection between medical chemistry and deep learning. Remarkable progress has been made in various applications on small molecules, ranging from generation (Jin et al., 2018; You et al., 2018) and property prediction (Gilmer et al., 2017; Cho & Choi, 2019; Klicpera et al., 2020) to protein-ligand interaction analysis (Lim et al., 2019; Wang et al., 2020), yet all these tasks rely on well-designed numerical representations, or *fingerprints*, of molecules. These fingerprints encode molecular structures and serve as the indicators in downstream tasks. Early work of molecular fingerprints (Morgan, 1965; Rogers & Hahn, 2010) started from encoding the two-dimensional (2D) structures of molecules, *i.e.* the chemical bonds between atoms, often stored as atom-bond graphs. More recently, a trend of incorporating molecular geometry into the representations arose (Axen et al., 2017; Cho & Choi, 2019).

Molecular geometry refers to the *conformation* (the three-dimensional (3D) coordinations of atoms) of a molecule, which contains widely interested chemical information such as bond lengths and angles, and thus stands vital for determining physical, chemical, and biomedical properties of the molecule. Whereas incorporating 3D geometry of molecules seems indeed beneficial, 3D fingerprints, especially in combination with deep learning, are still in infancy. The use of 3D fingerprints is limited by pragmatic considerations including i) calculation costs, ii) translational & rotational invariances, and iii) the availability of conformations, especially considering the generated ligand candidates in drug discovery tasks. Furthermore, compared with current 3D algorithms, mature 2D fingerprints (Rogers & Hahn, 2010; Gilmer et al., 2017; Xiong et al., 2020) are generally more popular with equivalent or even better performances in practice. For example, as a 2D approach, Attentive Fingerprints (Attentive FP) (Xiong et al., 2020) have become the *de facto* state-of-the-art approach.

To push the boundaries of leveraging 3D geometries in molecular fingerprints, we propose **HamNet (Molecular Hamiltonian Networks)**. HamNet simulates the process of molecular dynamics (MD)

---

\*Equal Contribution.
†Corresponding Author.

to model the conformations of small molecules, based on which final fingerprints are calculated similarly to (Xiong et al., 2020). To address the potential lack of labeled conformations, HamNet does not regard molecular conformations as all-time available inputs. Instead, A *Hamiltonian engine* is designed to *reconstruct* known conformations and *generalize for* unknown ones. Encoded from atom features, *implicit positions* and *momentums* of atoms interact in the engine following the discretized *Hamiltonian Equations* with learnable energy and dissipation functions. Final positions are supervised with real conformations, and further used as inputs to a Message-Passing Neural Network (MPNN) (Gilmer et al., 2017) to generate the fingerprints. Novel loss functions with translational & rotational invariances are proposed to supervise the Hamiltonian Engine, and the architecture of the Fingerprint Generator is elaborated to better incorporate the output quantities from the engine.

We show via our conformation-reconstructing experiments that the proposed Hamiltonian Engine is eligible to better predict molecular conformations than conventional geometric approaches as well as common neural structures (MPNNs). We also evaluate HamNet on several datasets with different targets collected in a standard molecular machine learning benchmark, MoleculeNet (Wu et al., 2017), all following the same experimental setups. HamNet demonstrates state-of-the-art performances, outperforming baselines including both 2D and 3D approaches.

## 2 PRELIMINARIES

**Notations.** Given a molecule with $n$ atoms, we use $\boldsymbol{v}_i$ to denote the features of atom $i$, and $\boldsymbol{e}_{ij}$ that of the chemical bond between $i$ and $j$ (if exists). Bold, upper-case letters denote matrices, and lower-case, vectors. All vectors in this paper are column vectors, and $\cdot^\top$ stands for the transpose operation. We use $\oplus$ for the concatenation operation of vectors. The positions and momentums of atom $i$ are denoted as $\boldsymbol{q}_i$ and $\boldsymbol{p}_i$, and the set of all positions of atoms in a molecule is denoted as $\boldsymbol{Q} = (\boldsymbol{q}_i, \cdots, \boldsymbol{q}_n)^\top$. $\mathcal{N}(v)$ refers to the neighborhood of node $v$ in some graph.

**Graph Convolutional Networks (GCNs).** Given an attributed graph $G = (V, \boldsymbol{A}, \boldsymbol{X})$, where $V = \{v_1, \cdots, v_n\}$ is the set of vertices, $\boldsymbol{A} \in \mathbb{R}^{n \times n}$ the (weighted) adjacency matrix, and $\boldsymbol{X} \in \mathbb{R}^{n \times d}$ the attribute matrix, GCNs (Kipf & Welling, 2017) calculate the hidden states of graph nodes as

$$\mathbf{GCN}^{(L)}(\boldsymbol{X}) \equiv \boldsymbol{H}^{(L)}, \quad \boldsymbol{H}^{(l+1)} = \sigma\left(\hat{\boldsymbol{A}}\boldsymbol{H}^{(l)}\boldsymbol{W}^{(l)}\right), \quad \boldsymbol{H}^{(0)} = \boldsymbol{X}, \quad l = 0, 1, \cdots, L-1. \quad (1)$$

Here, $\boldsymbol{H} = (\boldsymbol{h}_{v_1}, \cdots, \boldsymbol{h}_{v_n})^\top$ are hidden representations of nodes, $\hat{\boldsymbol{A}} = \boldsymbol{D}^{-\frac{1}{2}}\boldsymbol{A}\boldsymbol{D}^{-\frac{1}{2}}$ is the normalized adjacency matrix, $\boldsymbol{D}$ with $\boldsymbol{D}_{ii} = \sum_j \boldsymbol{A}_{ij}$ is the diagonal matrix of node degrees, and $\boldsymbol{W}$s are network parameters.

**Message-Passing Neural Networks (MPNNs).** MPNN (Gilmer et al., 2017) introduced a general framework of Graph Neural Networks (GNNs). In the $t$-th layer of a typical MPNN, *messages* $(\boldsymbol{m}^t)$ are generated between two connected nodes $(i, j)$ based on the hidden representations of both nodes $(\boldsymbol{h}^t)$ and the edge in-between. After that, nodes receive the messages and *update* their own hidden representations. A *readout function* is then defined over final node representations $(\boldsymbol{h}^T)$ to derive graph-level representations. Denoted in formula, the calculation follows

$$\boldsymbol{m}_v^{t+1} = \sum_{w \in \mathcal{N}(v)} M_t(\boldsymbol{h}_v^t, \boldsymbol{h}_w^t, \boldsymbol{e}_{v,w}), \quad \boldsymbol{h}_v^{t+1} = U_t(\boldsymbol{h}_v^t, \boldsymbol{m}_v^{t+1}), \quad \hat{\boldsymbol{y}} = R(\{\boldsymbol{h}_v^T | v \in V\}), \quad (2)$$

where $M_t, U_t, R$ are the *message*, *update* and *readout* functions.

**Hamiltonian Equations.** The Hamiltonian Equations depict Newton's laws of motion in the form of first-order PDEs. Considering a system of $n$ particles with positions $(\boldsymbol{q}_1, \cdots, \boldsymbol{q}_n)$ and momentums $(\boldsymbol{p}_1, \cdots, \boldsymbol{p}_n)$, the dynamics of the system follow

$$\dot{\boldsymbol{q}}_i \equiv \frac{\mathrm{d}\boldsymbol{q}_i}{\mathrm{d}t} = \frac{\partial \mathcal{H}}{\partial \boldsymbol{p}_i}, \quad \dot{\boldsymbol{p}}_i \equiv \frac{\mathrm{d}\boldsymbol{p}_i}{\mathrm{d}t} = -\frac{\partial \mathcal{H}}{\partial \boldsymbol{q}_i}, \quad (3)$$

where $\mathcal{H}$ is the *Hamiltonian* of the system, and equals to the total system energy. Generally, the Hamiltonian is composed of the *kinetic energy* of all particles and the *potential energy* as

$$\mathcal{H} = \sum_{i=1}^n \mathcal{T}_i + \mathcal{U}. \quad (4)$$

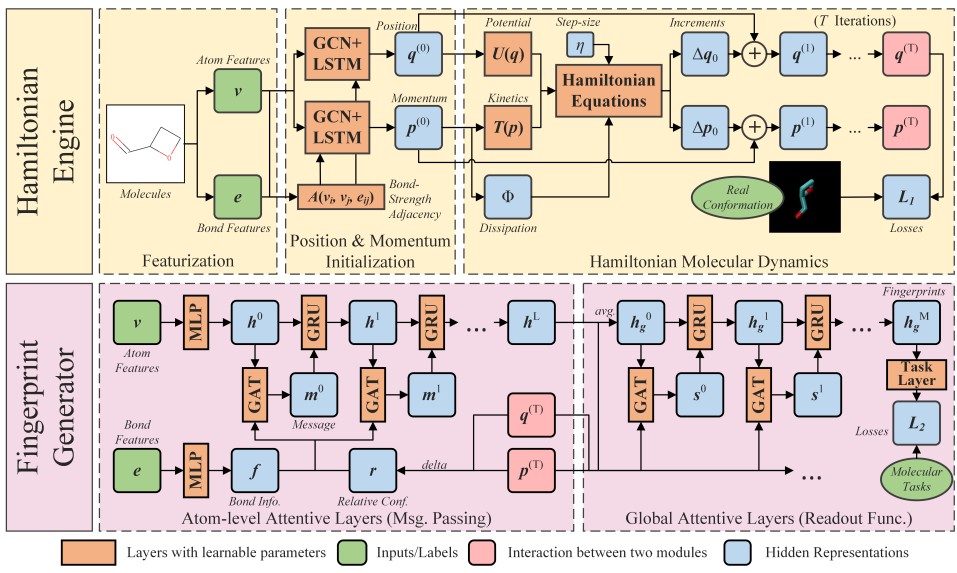

Figure 1: The overall structure of HamNet.

Meanwhile, if dissipation exists in the system, the Hamiltonian Equations shall be adapted as

$$\dot{\boldsymbol{q}}_i = \frac{\partial \mathcal{H}}{\partial \boldsymbol{p}_i}, \quad \dot{\boldsymbol{p}}_i = -\left(\frac{\partial \mathcal{H}}{\partial \boldsymbol{q}_i} + \frac{\partial \Phi}{\partial \dot{\boldsymbol{q}}_i}\right) = -\left(\frac{\partial \mathcal{H}}{\partial \boldsymbol{q}_i} + m_i \frac{\partial \Phi}{\partial \boldsymbol{p}_i}\right) \tag{5}$$

where $m_i$ is the mass of the particle and $\Phi$ is the *dissipation function* which describes how the system energy is dissipated by the outer environment.

## 3 METHOD: HAMNET

Figure 1 shows the overall architecture of HamNet. HamNet consists of two modules: i) a Hamiltonian Engine, where molecular conformations are reconstructed; and ii) a Fingerprint Generator, where final fingerprints are generated from atom & bond features and outputs from the Hamiltonian Engine.

### 3.1 HAMILTONIAN ENGINE

**Discretized Hamiltonian Equations.** The Hamiltonian Engine is designed to simulate the physical interactions between atoms in a molecule. To correctly incorporate the laws of motion, we discretize the Hamiltonian Equations with dissipation (Equation 5) and model the energy and dissipation with learnable functions. At the $t$-th step ($t = 0, 1, \cdots, T$) in the engine, for atom $i$,

$$\boldsymbol{q}_i^{(t+1)} = \boldsymbol{q}_i^{(t)} + \eta \frac{\partial \mathcal{H}^{(t)}}{\partial \boldsymbol{p}_i^{(t)}}, \qquad \boldsymbol{p}_i^{(t+1)} = \boldsymbol{p}_i^{(t)} - \eta \left(\frac{\partial \mathcal{H}^{(t)}}{\partial \boldsymbol{q}_i^{(t)}} + m_i \frac{\partial \Phi^{(t)}}{\partial \boldsymbol{p}_i^{(t)}}\right). \tag{6}$$

Here, $\eta$ is a hyperparameter of step size which controls the granularity of the discretization, $m_i$ is the (normalized) mass of atom $i$, and $\mathcal{H}^{(t)}, \Phi^{(t)}$ are the learnable Hamiltonian and dissipation functions of $\boldsymbol{q}^{(t)}$ and $\boldsymbol{p}^{(t)}$: (superscripts skipped)

$$\mathcal{H} = \sum_{i=1}^n T_i(\boldsymbol{p}_i) + U(\boldsymbol{q}_i, \cdots, \boldsymbol{q}_n), \qquad \Phi = \sum_{i=1}^n \phi(\boldsymbol{p}_i). \tag{7}$$

It should be noted that in order to improve the expressional power of the network, we extend the concept of *positions* and *momentums* into implicit ones in a *generalized $d_f$-dimensional space, i.e.* $\boldsymbol{q}, \boldsymbol{p} \in \mathbb{R}^{d_f}, d_f > 3$. We will discuss how to supervise these implicit quantities, and will show the influences of the dimensionality in the experimental results. Nonetheless, we parameterize the energy

and dissipation function in a physically inspired manner: for the atom-wise kinetic energy, we generalize the definition of kinetic energy ($T = \frac{p^2}{2m}$) as the *quadratic forms* of the implicit momentums; for the atom-wise dissipation, we adapt the *Rayleigh's dissipation function* ($\Phi = \frac{1}{2} \sum_{i,j=1}^n c_{ij} \dot{q}_i \dot{q}_j$) into the generalized space. Denoted in formula, we have

$$T_i(\boldsymbol{p}_i) = \frac{\boldsymbol{p}_i^\top \boldsymbol{W}_T^\top \boldsymbol{W}_T \boldsymbol{p}_i}{2m_i}, \qquad \phi_i(\boldsymbol{p}_i) = \frac{\boldsymbol{p}_i^\top \boldsymbol{W}_\phi^\top \boldsymbol{W}_\phi \boldsymbol{p}_i}{2m_i^2}. \tag{8}$$

For the potential energy, we simplify the *Lennard-Jones potential* ($U(r) = \epsilon \left(r^{-12} - r^{-6}\right)$) with parameterized distances $r$s in the generalized space, that is,

$$U = \sum_{i \neq j} u_{ij}, \quad u_{ij} = r_{ij}^{-4} - r_{ij}^{-2}, \quad r_{ij}^2 \equiv r(\boldsymbol{q}_i, \boldsymbol{q}_j) = (\boldsymbol{q}_i - \boldsymbol{q}_j)^\top \boldsymbol{W}_U^\top \boldsymbol{W}_U (\boldsymbol{q}_i - \boldsymbol{q}_j). \tag{9}$$

In both Equation 8 and Equation 9, $\boldsymbol{W}$s are network parameters. For the mass of the atoms, we empirically normalize the *relative atomic mass* $M_i$ of atom $i$ with $m_i = M_i/50$. One would also note that the parameters in each step of the engine remain the same, and thus the scale of parameters in the engine depends only on $d_f$ and is irrelevant to the depth $T$.

**Initial positions and momentums.** Graph-based neural networks are used to initialize these spatial quantities. Molecules are essentially graphs, while bonds between atoms can be of different types (single, double, triple and aromatic). Instead of using channel-wise GCNs which assign different parameters for different types of edges (Schlichtkrull et al., 2018), we calculate a *bond-strength adjacency* for the molecules: the strength of a bond depends on the *atom-bond-atom* tuple, *i.e.*

$$\boldsymbol{A}_{ij} = \text{sigmoid}\left(\textbf{MLP}\left((\boldsymbol{v}_i \oplus \boldsymbol{e}_{ij} \oplus \boldsymbol{v}_j)\right)\right) \text{ if the bond exists,} \quad \boldsymbol{A}_{ij} = 0 \text{ otherwise.} \tag{10}$$

With so-defined adjacency, we first encode the atoms with vanilla GCNs (Equation 1) and concatenate all hidden layers following the *DenseNet* scheme (Huang et al., 2017). Deep enough GCNs do capture entire molecular structures, however, atoms with identical chemical environment cannot be distinguished, for example, carbon atoms in the benzene ring. This may not be a problem in conventional MPNNs, but atoms with coinciding positions are inacceptable in physics as well as the Hamiltonian engine. Therefore, we conduct an $LSTM$ over the GCN outputs to generate unique positions for atoms. The order that the atoms appear in the LSTM conforms with the SMILES representations (Weininger, 1988) of the molecule, which display atoms in the molecule in a specific topological order. Denoted in formula, initial positions and momentums of atoms are calculated as

$$\tilde{\boldsymbol{q}}_i = \bigoplus_{l=0}^L \boldsymbol{f}_i^{(l)}, \qquad \boldsymbol{q}_i^{(0)} = \textbf{LSTM}_{s_i}\left(\tilde{\boldsymbol{q}}_{s_1}, \tilde{\boldsymbol{q}}_{s_2}, \cdots, \tilde{\boldsymbol{q}}_{s_n}\right) \tag{11}$$

$$\tilde{\boldsymbol{p}}_i = \bigoplus_{l=0}^L \boldsymbol{g}_i^{(l)}, \qquad \boldsymbol{p}_i^{(0)} = \textbf{LSTM}_{s_i}\left(\tilde{\boldsymbol{p}}_{s_1}, \tilde{\boldsymbol{p}}_{s_2}, \cdots, \tilde{\boldsymbol{p}}_{s_n}\right) \tag{12}$$

where $\boldsymbol{f}_i^{(l)}, \boldsymbol{g}_i^{(l)}$ are hidden representations of atom $i$ in the $l$-th GCN layer and $s_k$ is the atom at $k$-th position in the SMILES. As unique orders are assigned for atoms, their initial positions are thus unique.

**Conformation preserving.** After the dynamical process in the Hamiltonian Engine, positions in the generalized $\mathbb{R}^{d_f}$ space are transformed into real 3D space linearly, that is,

$$\hat{\boldsymbol{Q}} = \boldsymbol{Q} W_{trans}, \quad \hat{\boldsymbol{Q}} \in \mathbb{R}^{n \times 3}, \quad \boldsymbol{Q} = (\boldsymbol{q}_1, \cdots, \boldsymbol{q}_n) \in \mathbb{R}^{n \times d_f}. \tag{13}$$

Considering translational & rotational invariances, we do not require that $\hat{\boldsymbol{Q}}$ approximates the labeled 3D coordinations of atoms. Instead, three translational- and rotational-invariant metrics are proposed and used to supervise $\hat{\boldsymbol{Q}}$:

*I) Kabsch-RMSD (K-RMSD).* We use the *Kabsch Algorithm* (Kabsch, 1976) to rotate and align the approximated atom positions ($\hat{\boldsymbol{Q}}$) to the real ones ($\boldsymbol{Q}^R$), and then calculate the Root of Mean Squared Deviations (RMSD) of two conformations using atom mass as weights:

$$\hat{\boldsymbol{Q}}^K = \textbf{Kabsch}(\hat{\boldsymbol{Q}}; \boldsymbol{Q}^R), \quad L_{k-rmsd}(\hat{\boldsymbol{Q}}, \boldsymbol{Q}^R) = \sqrt{\frac{\sum_{i=1}^n m_i \times \left\|\hat{\boldsymbol{q}}_i^K - \boldsymbol{q}_i^R\right\|_2^2}{\sum_{i=1}^n m_i}}. \tag{14}$$

One should note that the alignment $\mathbf{Kabsch}(\cdot;\cdot)$ is calculated with SVD and is thus differentiable.

*II) Distance Loss.* Pair-wise distances between atoms are vital quantities in describing molecular conformations and enjoy desired invariances. Therefore, we propose the metric $L_{dist}$ as

$$L^2_{dist}(\hat{\boldsymbol{Q}}, \boldsymbol{Q}^R) = \frac{1}{n^2} \sum_{i,j=1}^{n} \left( \|\boldsymbol{q}_i - \boldsymbol{q}_j\|_2^2 - \|\boldsymbol{q}_i^R - \boldsymbol{q}_j^R\|_2^2 \right)^2 \tag{15}$$

*III) ADJ-k Loss.* A drawback of the naive distance loss is that distances between far atoms are over-emphasized, leading to deformed local structures. Therefore, we further propose *ADJ-k loss*, where only distances between $k$-hop connected atoms are preserved under weights calculated from hop-distances. Denote the *normalized*, *simple* adjacency matrix as $\tilde{\boldsymbol{A}}$,[1] the ADJ-k loss is defined as

$$L^2_{adj-k}(\hat{\boldsymbol{Q}}, \boldsymbol{Q}^R) = \frac{1}{n} \sum_{i,j=1}^{n} \tilde{\boldsymbol{A}}_{ij}^k \left( \|\boldsymbol{q}_i - \boldsymbol{q}_j\|_2^2 - \|\boldsymbol{q}_i^R - \boldsymbol{q}_j^R\|_2^2 \right)^2 \tag{16}$$

In implementation, we use a linear combination of *K-RMSD* and *ADJ-3* losses to supervise the engine, *i.e.* $L_{HE} = L_{k-rmsd} + \lambda L_{adj-3}$, where $\lambda$ is a hyperparameter.

## 3.2 FINGERPRINT GENERATOR

After the dynamical process in the Hamiltonian Engine, the molecular fingerprints are generated with the outputs as well as atom & bond features. The architecture of the Fingerprint Generator can be seen as an MPNN (Gilmer et al., 2017) instance. Analogous to that in Attentive FP (Xiong et al., 2020), *messages* are generated with Graph Attention Layers (GAT) (Velickovic et al., 2018), and hidden representations of nodes are *updated* with Gated Recurrent Units (GRU) (Cho et al., 2014). Nonetheless, the architecture of HamNet is further adapted as conformation-aware: we modify the calculation of messages and attentive energies to incorporate relative positions and momentums. Denoted in formula, the atom-level calculation in the Fingerprint Generator follows

$$\boldsymbol{h}_i^0 = \mathbf{MLP}(\boldsymbol{v}_i), \quad \boldsymbol{f}_{ij} = \mathbf{MLP}(\boldsymbol{e}_{ij}), \quad \boldsymbol{r}_{ij} = (\boldsymbol{q}_i \oplus \boldsymbol{p}_i) - (\boldsymbol{q}_j \oplus \boldsymbol{p}_j); \tag{17}$$

$$\epsilon_{ij}^l = (\boldsymbol{w}_\epsilon^l)^\top (\boldsymbol{f}_{ij} \oplus \boldsymbol{r}_{ij}), \quad \alpha_{ij}^l = \mathrm{softmax}(\{\epsilon_{ij}^l | j \in \mathcal{N}(i)\}), \tag{18}$$

$$\boldsymbol{m}_i^{l+1} = \sum_{j \in \mathcal{N}(i)} \alpha_{ij}^{l+1} \boldsymbol{W}_M^l \left[ \boldsymbol{h}_i^l \oplus \boldsymbol{r}_{ij} \oplus \boldsymbol{h}_j^l \right], \quad h_i^{l+1} = \mathbf{GRU}(\boldsymbol{h}_i^l, \boldsymbol{m}_i^{l+1}), \quad l = 0, 1, \cdots, L. \tag{19}$$

Atom representations in the last layer ($h_i^L$s) then serve as inputs to a *global attentive readout function*. A virtual *meta node* ($g$) is established and connected to all atoms in order to conduct $M$ layers of attentive pooling. Similar to that in atom-level calculation, the positions and momentums of atoms are incorporated in the calculation:

$$\boldsymbol{h}_g^0 = \frac{1}{n} \sum_i \boldsymbol{h}_i^L; \quad \eta_i^m = \left[ \boldsymbol{h}_g^l \oplus \boldsymbol{q}_i \oplus \boldsymbol{p}_i \oplus \boldsymbol{h}_i^L \right], \quad \beta_i^m = \mathrm{softmax}(\{\eta_i^m | i \in V\}), \tag{20}$$

$$\boldsymbol{s}_g^m = \sum_i \beta_i^m \boldsymbol{W}_s^m \left[ \boldsymbol{q}_i \oplus \boldsymbol{p}_i \oplus \boldsymbol{h}_i^L \right], \quad \boldsymbol{h}_g^{m+1} = \mathbf{GRU}(\boldsymbol{h}_g^m, \boldsymbol{s}_g^m), \quad m = 0, 1, \cdots, M. \tag{21}$$

Here, $\boldsymbol{s}_g$ is the *global message*, and $V$ is the set of all atoms. The final output, $h_g^M$, is the desired fingerprints of the target molecules. The same as current neural molecular fingerprints (Duvenaud et al., 2015; Xiong et al., 2020), the generated fingerprints are then supervised with molecular properties, such as regression and classification tasks.

---

[1]The simple adjacency matrix refers to the indicator matrix of bond existences, regardless of bond types; the normalization is conducted the same as $\hat{A}$ in GCNs (See Section 2).

### 3.3 DISCUSSION

From a physical perspective, the Hamiltonian Engine is essentially inspired by and highly related to Molecular Dynamics (MD). The potential energy function in the engine can be regarded as a generalized while simplified molecular force field. Force fields are widely used tools in molecular simulation, where potential energies are modeled as a family of functions of conformations, whose parameters are calculated with quantum chemistry or determined by experiments. After a force field is established, conformations are optimized by minimizing the potential energy. Similarly, the dissipation we introduced in the Hamiltonian Engine serves as an implicit optimization of potential energy, as the system energy is continuously dissipated through $\Phi$ during the dynamical process. As a result, after adequate steps, the molecular conformations always converge to a local minimum of the potential energy, and the momentums of atoms converge to 0. From a deep learning perspective, the Hamiltonian Engine can be seen as a pair of dual, residual MPNNs operating on fully-connected graphs: at each step, messages calculated by $q$ influence $p$ and *vice versa*. Under this view, the most essential difference of introducing physical laws is that the messages passed from $q$ to $p$ is *symmetric* between any two given atoms, following the *Newton's third law* and conforming a *conservative field* (the potential force field). Speaking more detailly, *$q$-messages* sent between a pair of atoms (*i.e.* the forces, $\partial \mathcal{H}/\partial q$) are implicitly guaranteed as symmetric, yet the actual *$p$-update* (*i.e.* the accelerations, $\dot{p}/m$) may not be: they are also related to properties of the receiver (the atom mass) and the outer environment (the dissipation).

## 4 RESULTS

### 4.1 EXPERIMENTAL SETUP

**Datasets.** Five molecular datasets are used to evaluate HamNet, including a *Quantum Mechanics* dataset (`QM9`) and four *biomedical* datasets, namely `Tox21`, `Lipop`, `FreeSolv`, and `ESOL`. QM9 (Ramakrishnan et al., 2014) contains calculated conformations and 12 quantitative quantum-chemical properties of 133k molecules; `Tox21` contains 12 binary toxicological indices of $7,831$ molecules; `Lipop` contains quantitative liposolubility lipophilicity of $4,200$ molecules; `FreeSolv` contains quantitative hydration free energy of $642$ molecules, and `ESOL` contains quantitative solubility of $1,128$ molecules. All datasets are referred in MoleculeNet, and the same metrics [2], data split ratios [3], and multi-task scheme (for `QM9` and `Tox21`) [4] are used in our paper.

**Featurization and Implementation.** We use the identical featurization as Attentive FP (Xiong et al., 2020). In total, 39-dimensional atom features (including atom types, atom degree, indicators of aromaticity and chirality *et al*) and 10-dimensional bond features (including bond types, indicator of conjugation *et al*) are derived from the molecules. One could refer to the Appendix for more details of featurization. As a default setup, we use a 20-step ($T = 20$) Hamiltonian Engine with $d_f = 32$, and $L = 2, M = 2$ with 200-dimensional hidden representations ($\dim(h_i) = \dim(h_g) = 200$) in the Fingerprint Generator. For the training of HamNet, we first train the Hamiltonian Engine with known conformations,[5] and use the output to train the Fingerprint Generator, with *mean-squared-error* losses for regression tasks with RMSE metric, *mean-absolute-error* losses for those with MAE metric, and *cross-entropy* losses for classification tasks. Other implementation details, including the choices of hyperparameters on different datasets and the training setup are available in the Appendix.

### 4.2 CONFORMATION PREDICTION

We evaluate the ability of the Hamiltonian Engine in predicting molecular conformations on `QM9`, where known conformations of molecules are available, and reported the *Kabsch-RMSD $L_{k-rmsd}$* and *distance loss $L_{dist}$* losses. Two baselines are compared against the Hamiltonian Engine (*Ham. Eng.*): i) an MPNN with the exact architecture proposed in (Gilmer et al., 2017), supervised in the

---

[2]We use MAE for `QM9`, ROC for `Tox21`, and RMSE for `Lipop`, `FreeSolv` & `ESOL`.

[3]Data are randomly split to $8 : 1 : 1$ as training, validation and test sets.

[4]Models are trained to simultaneously preserve all targets after standard normalization, and averaged performances are reported.

[5]On datasets without known conformations, we generate labeled conformations with RDKit to train the engine

Table 1: Quantitative results of conformation prediction on QM9.

| METRIC | Kabsch-RMSD ($\mathring{A}$) | Distance Loss ($10^{-2}\mathring{A}$) |
|---|---|---|
| MPNN | 1.708 | 8.620 |
| RDKit | 1.649 | 7.519 |
| Ham. Eng. (*w/o LSTM*) | 2.039 | 10.871 |
| Ham. Eng. (*w/o dyn.*) | 1.442 | 5.519 |
| Ham. Eng. (*w/o* $\Phi$) | 1.389 | 5.227 |
| Ham. Eng. (*w/o ADJ-3*) | **1.084** | 7.746 |
| Ham. Eng. (*as proposed*) | 1.384 | **5.186** |

Figure 2: Visualized conformations at different steps of the Hamiltonian Engine.

same way as we have introduced in Section 3.1; ii) a Distance Geometry (Blaney & Dixon, 2007) method tuned with the *Universal Force Field* (UFF), implemented in the RDKit package (referred as *RDKit*) [6]. We also conduct an *ablation analysis* of the Hamiltonian Engine by testing: i) an engine with the LSTM removed (*w/o LSTM*); ii) an engine with no Hamiltonian dynamics, *i.e.* $T = 0$ (*w/o dyn.*); iii) an engine without the dissipation function (*w/o* $\Phi$); and iv) an engine trained without the ADJ-3 loss (*w/o ADJ-3*).

Table 1 shows the two losses on the test sets. Our approach outperforms the Distance Geometry baseline (RDKit) by 16%, while the MPNN cannot. In the ablation analysis, effectiveness of different components is also varified: improvements on both metrics are observed when *LSTM*, *molecular dynamics*, and *dissipation function* exist. Although training the Hamiltonian Engine simply with Kabsch-RMSD leads to better performances on the very metric, distance losses of these models are unacceptably large. This indicates although atoms tend to appear closer to their labeled locations, the relative structures inside the molecules are compromised, which is a particularly undesired result in molecular science. One could refer to Figure 2 for a more intuitive understanding of the dynamics in the Hamiltonian Engine: atoms in the initial conformations ($\hat{Q}^{(0)}$) tend to gather around the molecular centers, and the repulsion forces derived from the potential energy *stretch* the molecules into the correct conformations ($Q^R$). As dissipation exists in the system, conformations converge to the real ones after adequate steps.

## 4.3 MOLECULAR PROPERTY PREDICTION

We compare HamNet with five baselines on the molecular property prediction tasks. i) *MoleculeNet* (Wu et al., 2017) tested a collection of molecular representation approaches at the time, by which we present the best performances achieved. ii) *3DGCN* (Cho & Choi, 2019) augmented conventional GCN-based methods with input bond directions. iii) *DimeNet* (Klicpera et al., 2020) proposed directional message passing where messages instead of atoms are embedded. iv) *Attentive FP* (Xiong et al., 2020) proposed an MPNN instance where local and global attentive layers are used. v) *CMPNN* (Song et al., 2020) strengthened the message interactions between nodes and edges through a communicative kernel. Two HamNet variants are also tested: i) a HamNet without known conformation, where all $q$, $p$-related components in the Fingerprint Generator are removed;

---

[6]We use the 2020.03.1.0 version of the RDKit package. See `http://www.rdkit.org/`

Table 2: Quantitative results on various datasets of baselines, HamNet, and its variants. Baselines using 3D conformations of test molecules are marked *italic*. For different metrics, "↑" indicates that the higher is better, "↓" contrarily. We directly take reported performances from corresponding references, and leave unreported entries blank ("–").

| DATASET METRIC | QM9 Multi-MAE↓ | Tox21 Multi-ROC↑ | Lipop RMSE↓ | FreeSolv RMSE↓ | ESOL RMSE↓ |
|---|---|---|---|---|---|
| MoleculeNet (2017) | 2.350 | 0.829 | 0.655 | 1.150 | 0.580 |
| *3DGCN* (2019) | – | – | – | 0.824±0.014 | 0.558±0.069 |
| *DimeNet* (2020) | 1.920 | – | – | – | – |
| Attentive FP (2020) | 1.292 | 0.857 | 0.578 | 0.736 | 0.505 |
| CMPNN (2020) | – | 0.856± 0.006 | – | 0.808±0.129 | 0.547±0.011 |
| HamNet (*w/o conf.*) | 1.237±0.030 | 0.868±0.012 | 0.572±0.011 | 0.840±0.023 | 0.547±0.015 |
| *HamNet (real conf.)* | 1.199±0.017 | 0.864±0.006 | 0.566±0.015 | 0.811±0.048 | 0.584±0.012 |
| HamNet (*ours*) | **1.194±0.038** | **0.875± 0.006** | **0.557±0.014** | **0.731±0.024** | **0.504±0.016** |

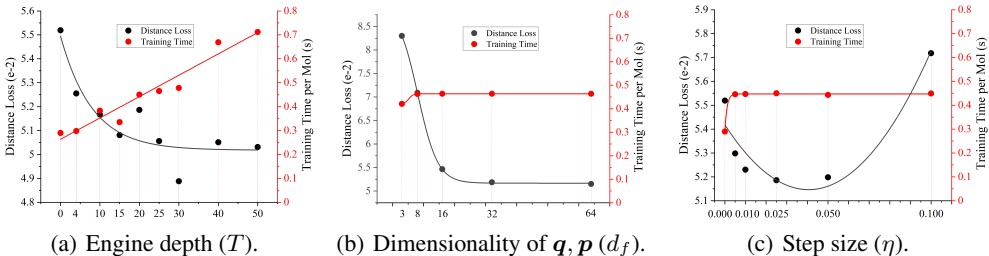

(a) Engine depth ($T$).      (b) Dimensionality of $\boldsymbol{q}, \boldsymbol{p}$ ($d_f$).      (c) Step size ($\eta$).

Figure 3: Effects on conformation prediction of hyperparameters in the Hamiltonian Engine. Distance losses and running time versus **(a)** the engine depth $T$; **(b)** the dimensionality of the generalized space $d_f$; and **(c)** the step size $\eta$ of discretization are plotted.

ii) a HamNet with real conformations, where the Hamiltonian Engine is removed and all $\boldsymbol{q}, \boldsymbol{p}$s are transformed from real coordinations, with MLPs, to the corresponding dimensionality. Five replicas of HamNet models are trained with means and standard deviations reported.

Table 2 shows the quantitative results of the fingerprints. As HamNet and all chosen baselines follow the same evaluation scheme proposed in MoleculeNet, we directly present the reported performances and leave unreported ones blank. We also *italicize* baselines which leverage true 3D conformations of the test set. HamNet is able to significantly outperform all baselines on all datasets. Moreover, compared with the HamNet variant using real conformations, HamNet with the Hamiltonian Engine still performs better. We believe the reason is that as the Hamiltonian Engine is trained with translation- & rotation-invariant losses, the generalized space enjoys more robustness compared with real coordinations of atoms used directly.

## 4.4 PARAMETER ANALYSIS

We conduct further analysis of the effects of several important hyperparameters on the conformation prediction performances in the Hamiltonian Engine, including the engine depth ($L$), the dimensionality of the generalized space ($d_f$), and the step size ($\eta$). Figure 3 demonstrate the distance loss and / or the running time. i) As the depth increases, the running time increases linearly, and the test loss gradually decreases until convergence. Empirically, 25-30 steps suffice. ii) The running time of the engine is hardly influenced by the dimensionality $d_f$, while the performance enjoys a significant improvement by increasing the dimensionality when $d_f \leq 32$. iii) An appropriate choice of the step size is crucial. For example, with the fixed engine depth $T = 20$, an ideal choice of the step size would be $\eta \in [0.025, 0.050]$.

## 5 DISCUSSION & FUTURE WORK

In this paper, we proposed a novel molecular representation approach, Molecular Hamiltonian Network (HamNet). Instead of directly using conformations as inputs, HamNet learns to predict real conformations with a physically inspired module, the Hamiltonian Engine. Novel loss functions with translational & rotational invariances are proposed to train the engine. Final representations of molecules are generated with a Fingerprint Generator, whose architecture is based on MPNNs and considerately modified to incorporate generated implicit conformations. We discussed the relationships between physics and deep learning inside the Hamiltonian Engine from both perspectives, and we believe that the physics-based model enjoys better interpretability than general MPNNs. We further demonstrated with our experiments that the proposed Hamiltonian Engine is eligible to learn molecular conformations, and that HamNet achieves state-of-the-art performances on molecular property prediction tasks in a standard benchmark (MoleculeNet).

It should be noted that a recent trend in incorporating machine learning, especially deep learning, into modeling molecular potentials emerged (Chmiela et al., 2017; 2018; Zhang et al., 2018). Another related field of HamNet is neural physics engines (Sanchez-Gonzalez et al., 2018; 2019; Greydanus et al., 2019), which learn to conduct simulations that conform to physical laws. The design of the Hamiltonian Engine in our paper is highly motivated by these works, while the ultimate goal of HamNet is to derive well-designed molecular representations, instead of to accurately model the molecular dynamics. Also, instead of using the structural optimization to derive stablized conformations after a force-field (potentials in HamNet) is established, HamNet uses the Hamiltonian Engine to make the whole process differentiable.

For future work, a promising aspect would be to elaborate the learnable potential, kinetics and dissipation functions used in the Hamiltonian Engine. Work in using HamNet in more straight-forward applications would also be useful, such as virtual screening, protein-ligand binding prediction, *etc*. In addition, an interesting attempt in further modifying HamNet would be to change the current discretization approach of the Hamiltonian Equations to Neural ODEs (Chen et al., 2018; Sanchez-Gonzalez et al., 2019), which may yield a finer-grained simulation of the molecular dynamics.

### ACKNOWLEDGMENTS

This work was supported by the National Natural Science Foundation of China (Grant No. 61876006). We would also like to thank Dr. Chenbo Wang for his help in the physics theories of this paper.

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
