# OpenReview forum: "Conformation-Guided Molecular Representation with Hamiltonian Neural Networks"
_ICLR.cc/2021/Conference — ICLR 2021 Poster_

### Official Review · AnonReviewer3 · 2020-10-28
**Official Blind Review #3**

**Rating:** 7
**Confidence:** 3

**Review:**

**Summary**
The paper proposes a new method for generating fingerprints for small molecules. It is based on two components: a "Hamiltonian Engine" that runs a brief simulation, predicting the structure of the small molecule by minimizing a learned potential energy, and 2) a message passing algorithm that uses the predicted structure as input. The reported experimental results demonstrate state-of-the-art performance.

**Strengths**
 o The manuscript addresses an important problem
 o The Hamiltonian Engine is a non-trivial contribution, which should be of broad interest.
 o The reported results seem to significantly outperform the current state of the art

**Weaknesses**
The paper is information-dense and in some places difficult to read.

**Recommendation**
I recommend this paper be accepted. To my knowledge, the presented method is novel, and the results are convincing. Furthermore, it is an interesting contribution on the interface between physics and machine learning, which should be of interest even beyond the small-molecule ML subfield. The manuscript would however benefit from a slightly clearer presentation (see below).

**Detailed feedback**

Although I'm generally positive about the article, there are several places where I would recommend sentences could be improved to more clearly communicate the central ideas in the paper:

Page 1. The last paragraph on page 1 should make it clearer what the role of the Hamiltonian Engine is. In my opinion, the choice of the word "preserve", is particularly confusing - it seems to me more like a "prediction" or a "reconstruction". The work "preserve" made me think that the positions from the original molecule were used to initialize the positions and momenta of the simulator ("preserve" thus meaning the ability of the forcefield to stabilize a given structure). Figure 2 cleared this up for me, but that does not appear until page 7. Also, the authors write that HamNet "simulates the process of molecular dynamics", but it is not clear why  a Hamiltonian Engine is needed at all - couldn't you simply have used a short MD simulation with an established molecular force field?

Figure 1. Could anything be done visually to show how the Fingerprint Generator uses the output from the Hamiltonian Engine? Currently, the two methodological components of the paper seem a bit disconnected in the figure.

Page 4. "Graph-based neural networks are used to initialize these spatial quantities." It would be informative if you could add a few lines to motivate this choice. Couldn't you have used some idealized positions as a starting point for your simulation? Later on in the paper you mention that you use RDKit to create positions when they are not available in the data - couldn't you then just always use RDKit to create an initial structure - or would this be less accurate or too slow?

Page 5. "conformations always converge to a local minimum" Perhaps you could comment on the nature of the learned energy landscape.  Do we expect it to be a highly multimodal landscape, with risk for convergence to suboptimal local optima? Is this why you place so much emphasis on the initialization of your simulations?

**Clarifying questions to the authors**
Page 4. "we simplify the Lennard Jones potential"
Why do you make the simplification to r^{-4} - r^{-2}? It seems to me that you could have calculated the standard r^{-12} - r{^-6} equally well from the r^2 that you have available.

Page 4. "we empirically normalize the relative atomic mass".
What is the reason for this normalization? It it just to improve the training behavior of the model? Is this a hyperparameter that needs to be optimized?

Figure 2. Does the Step 0 correspond to an initialization produced by the GCN+LSTM? The positions look very compact - wouldn't you have expected them to look more realistic?

Page 6. "tuned with the Universal Force Field" Could you have done a distance geometry initialization but using your learned forcefield for subsequent fine-tuning?. Would this be superior to using the Universal Force Field?

Page 7, "where the Hamiltonian Engine is removed and all q,ps are transformed from real coordinations"
If you are only given structures, how to you obtain the momenta?

Page 8. I found it interesting that it was necessary to map the positions and momenta to a higher dimensional space for optimal performance. Do you know if this overparamterization is necessary purely to facilitate gradient-based training, or whether there are more fundamental reasons for modelling them this way?

**Minor comments**
Page 1. "For examples, Attentive Fingerprints have yet become the de facto state-of-the-art approach".
To a reader not intimitely familiar with these references it is not clear what you mean by this sentence. Is Attentive FP a 2D method? I think there is also a "to" missing before "become".

Page 4. "but atoms with coincide positions". "coincide" -> "coinciding"

Page 4, "we do not explicitly require \hat Q to conform to labeled coordinations of atoms". This was a bit difficult to read. I assume you mean that you only predict Q up to a rotation and a translation. Perhaps rephrase.

Page 5. "Speaking for detailly". Rephrase.

---

> ### Author Response · Authors · 2020-11-22
> **Official reply to Reviewer 3**
>
> We truly appreciate your recommendation for our paper, and all the valuable comments and detailed suggestions. With your help, we revised some figures and sentences in our paper. To conclude:
>
> 1. We improved Figure 1 to better demonstrate the in/output of the two modules and the data flow in between;
>
> 2. We used "reconstruct" instead of "preserve" to clear the confusion;
>
> 3. We rephrased some of the typos / confusing statements follow your "Minor comments" section.
>
> With regard to your questions, we would like to clarify:
>
> 4. **Initialization** (why use GNN(+LSTM) to initialize and not start with real confs; why Step 0 looks compact; why a higher dimensional space is in need; and eventually, why Ham. Eng. at all?)
>
> Firstly, as we claimed in the paper, real conf (or RDKit simulated ones) are not always available. We do use RDKit confs when the datasets lack real ones, but they are calculated *only for the training set*. Actually, in real scenarios, confs of training sets are generally available; if not, at least the calculation is once-and-for-all. What HamNet saves is the time of the conf calculation in the *inference stage*, where labeled confs are NOT used. In the scenarios including drug candidate generation & large-scale virtual screening, this improvement is crucial.
>
> Secondly, we would like to stress that preserving the conformations is not the ultimate goal for HamNet, but to *derive good representations* is. Therefore, the desired properties of the initializations are not quite the same as classical MD. Starting from 2D structures & atom / edge features with GNN encoders enables the Ham. Eng. to be not only *spatial-aware* (supervised with real confs), but also *structural- & featural-aware*. In another word, the role Ham. Eng. plays is not only to derive good conformations, but also to merge the structural & featural information (input) with the spatial information (output) in the generalized space. We believe this also answers the question **why a higher dimensional space is in need**.
>
> Experimental results also support the advantage of this setup: in the experiments *HamNet (real conf.)* (Table 2), we tested the results of directly using real confs. as inputs (3D coordinations of atoms mapped with MLP to $2 \times d$ dimensional to eliminate the effect of dimensionality), while no better results than *HamNet (ours)* are obtained. We believe this proves the effects of structural- & featural info in the generalized space.
>
> Initialization looking compact is thus comprehendable: they are encoded from atom / edge features directly, and as the GNN+LSTM is regularized, the outputs of them approaching 0 is in the expectation. Note that we only poses conformation supervision on the final outputs, and the initializations are not explicitly scaled: it is done by the Ham. Eng.. The process inside the Ham. Eng. attempts to "stretch" the feature-based conformation initializations to correct conformations. In this process, the mergence of two types of information is done.
>
> 5. **Why simplifying the LJ potential?**
>
> The function $r^{-12}-r^{-6}$ is indeed calculatable, however, given the circumstances that they appear in a (somehow complicated) neural network, using high orders of polynomials may lead to computational issues such as gradient explosion / vanishing et al, and experimentally, it does not work well. The polynomial $r^{-4} - r^{-2}$ has a similar landscape with lower order, so we made the simplification.
>
> 6. **Normalization of mass**
>
> Normalization of mass is for computational convenience. Note that in quadratic functions $T(p)$ and $U(q)$, linear scales can be adaptively adjusted by the *singular values* of the weight matrics $W_T,W_U$, e.g.
> $$
> \frac{p^TW_T^TW_Tp}{2 \alpha m} = \frac{p^T(\alpha^{-1/2}W_T)^T(\alpha^{-1/2}W_T)p}{2m} := \frac{p^T\tilde{W}_T^T\tilde{W}_Tp}{2m}.
> $$
> Therefore, the scale of mass is not that important, as long as it is computational convenient. Using $m=M/50$ is empirically appropriate as $m \in (0.1, 1)$ (heavy atoms).
>
> 7. **Initialization of the momenta**
>
> We initialize the momenta in the same way as the positions, i.e. using GNN+LSTM. Indeed, this does not lead to direct interpretability as drawing from a Boltzmann distribution does; however, as all the interaction happens in a generalized space, we leave the neural networks to do their job.

---

### Official Review · AnonReviewer2 · 2020-10-28
**A very interesting work, but with a lot of questions from the unclear presentation.**

**Rating:** 7
**Confidence:** 5

**Review:**

Summary: This paper presents a novel neural network module called Hamiltonian Neural Networks to learn representation of molecules. The module consists of two main components: 1) Hamiltonian Engine (HE), and 2) Fingerprint Generator (FG). The HE 1) takes a molecular graph with atom and bond features as inputs, and first generate "generalized" positions p and momentums q via GNN+LSTM. These ps and qs are fed into a discrete Hamiltonian system with dissipation, and produces (generalized) "conformations" of molecules. The FG also takes a molecular graph with atom and bond features + the generalized positions and momentums from 1) as inputs to generate the final vector embedding of the input molecule. HE is trained to fit the input 3D conformation with a weighted combination of two types lossses (K-RMSD + 5 * ADJ-k loss). Posing this "Hamiltonian" inductive bias to the model, the paper demonstrated that the prediction performance over multiple molecular tasks from MoleculeNet are all improved. Also, the analysis of HE module or the hyperparamter sensitivity analysis are provided.

Comments:
This is a very interesting work in the light of molecular representation learning. The 3D geometries of molecules, i.e. conformations, are actually not rigid-body-like, rather has a degree of freedom varied according to the physical rule as we see them in molecular dynamics. It would be reasonable to see that this inductive bias explitly as "Hamiltonian system with learnable parameters" actually brings empirical performance improvements.

I enjoyed the paper and core ideas, but the description and presentation of the (complicated) method's inner workings would need to be improved. In particular, it'll be nice to make clear the following points:

- There are no description on the classifier or regressor head for downstream tasks of MoleculeNet, and how to train the entire network. This should be explicitly included in the paper. The paper only provides how to get the fingerprint, but seemingly there are no explanations or description on how to train this FG (or the entire module of HE+FG). We'll need to add some task-specifc heads after HE -> FG for MoleculeNet tasks. Then, is HE first trained with K-RMSD + 5 * ADJ-k loss, and the entire HE + FG fitted with the task-specific supervision loss? Or the entire HE + FG fitted with K-RMSD + 5 * ADJ-k loss + alpha * task-specific supervision loss (with some alpha)?

- The model desciprion of Figure 1 is unclear and rough, and it was quite painstaking to get the information on what are learnable parameters and what are just states or intermediates. Also, the p and q from HE to FG would be also hard to capture at first glance. I understand that the module is quite complicated and not easy to depict it in one figure, but I'll appreciate if the authors can include all learnable parameters in Figure 1. (For example, like the model figure, Fig.4, of the DimeNet paper).

- Related to the above problem, some "ablation study" would be quite unclear. For example, "HamNet (real conf.)" in Table 2 is to tweak HE somehow, or just feed the real 3D conformations into FG with momentums of 0s (with removing the entire HE processing)?? In the first place, the HE used 32-dimensional vector for ps and qs. Is it fair if this means to use 3-dimensional vector instead (at least, need to pass it to MLP for converting the same dimension?). What if we use 3-dimension for d_f??

- "HamNet (w/o conf.)", "Ham. Eng. (w/o LSTM)", "Ham. Eng. (w/o dyn.)" are also quite unclear.

- It is quite interesting to see "HamNet (ours)" was better than "HamNet (real conf)" even for QM9, and moreover, given that for most tasks, the 3D input conformation is generated by RDKit distance geometry with UFF. But the internal conformation of HamNet is actually "d_f"-dimensional (32-dimensional), and this difference could come from this expressiveness. To confirm this, it would be nice to include the performance of "HamNet (ours) with d_f = 3". (By the way, since the 2018.09 release of the RDKit, ETKDG is the default conformation generation method. What if we use this instead of the distance geometry...?)

- It would be very nice to add some remarks on related work for readers. The name and method would definitely remind related work such as "Hamiltonian Neural Networks (Greydanus et al, NeurIPS 2019)" , Machine-learning potential for MD (see [a][b] below, for example), and and "learning to simulate" methods. See the “delta graph network” (DeltaGN) baseline of the cited workshop paper by Sanchez-Gonzalez et al, 2019.

[a] Chmiela et al, Machine learning of accurate energy-conserving molecular force fields (Science Advances, 2017)
[b] Chmiela et al, Towards exact molecular dynamics simulations with machine-learned force fields (Nature Communications, 2018)

---

> ### Author Response · Authors · 2020-11-22
> **Official reply to Reviewer 2**
>
> We truly appreciate your recommendation for our paper. Meanwhile, we are sorry to cause the confusions that you noted in the comments. To address to your concerns, we would like to clarify:
>
> 1. **The training of HamNet**
>
> We are sorry to miss out the training details in the mainbody of the paper. With your help, we revised the paper and addressed the training scheme of HamNet in both Section 3.2 (FG) and Section 4.1 (implementation). Currently, we adopt a separate training strategy which first pretrain HE with loss $L=L_{k-rmsd} + 5 L_{adj-3}$, and then train the FG with outputs from HE. As for the training losses of FG, we use MAE for QM9, MSE for other regression tasks, and cross entropy for the classification tasks.
>
> We also tried to conduct joint training (i.e. train FG+HE for specific tasks), while no significant improvement was observed, the parameters in HE is hardly altered, and the complexity is rather high. Therefore, we report only the separate training strategy in our paper.
>
> 2. **Improving Figure 1**
>
> Indeed it is hard to plan the figure, and we thank you for the consideration. What we would like to show via Figure 1 is the major data flow in the model, while listing all parameters & details as DimeNet does in Figure 1 is too difficult, especially considering there are complicated architectures including GRUs and (modified) GATs.
>
> We revised Figure 1 and used different colors to mark out 1) the data interaction between two modules (red); 2) all layers with learnable parameters (orange); 3) inputs / outputs (labels) (green); 4) all other hidden representations (blue). We hope you'll find it clearer.
>
> 3. **Ablation study details**
>
> Due to the limitation of space, we did not provide detailed explanations in the mainbody of the paper. We hope that Section B in the Appendix helps to comprehend the setups. We also revised this part to become more clear about the  implementation.
>
> 4. **Experiments with $d_f = 3$**
>
> The results of conformation prediction of Ham. Eng. with $d_f = 3$ is available in Figure 3 (d), and we see that it is far from sufficient in capturing the molecular conformations. Therefore, we did not do experiments on HamNet with $d_f=3$. We would anticipate that HamNet ($d_f=3$) behaves similarly to HamNet (w/o conf), as $q,p$ fails to capture valuable information in the labeled conformations.
>
> 5. **More citations on ML (DL) potentials & neural physics engines**
>
> We sincerely agree that some discussion on ML potentials and neural physics engines is beneficial. In fact, HamNet is highly motivated by these works. However, due to the limitation of space, we could not display these discussions in our first submission (the paper is *information-dense* already). As the page limitation is relaxed in the rebuttal stage, we revised our submission and added corresponding contents in Section 5.

---

### Official Review · AnonReviewer1 · 2020-10-28
**Interesting approach for incorporate 3d inforamtion in molecular fingerprints, but lacking results**

**Rating:** 5
**Confidence:** 4

**Review:**

This paper proposes to use 3d conformations for learning molecular fingerprints by 1) training a generative model to predict the 3d coordinates and 2) use those to train a "fingerprint generator" to obtain fingerprints by learning to predict molecular properties. The paper is well structured and clearly written. The intuition behind the Hamiltonian engine is nicely explained in the discussion section. The idea of a two step procedure to include 3d structure information in the training even when it is not available at test time is interesting and original.

Unfortunately, some of the design decisions did not became quite clear to me: Why is the relaxation modeled with an MD-like dynamics approach instead of structure optimization, i.e. following the gradient of the energy? Although, here is an ablation study for leaving out the dynamics that performs worse, this might rather indicate that the initial guess of the positions is not good enough to find the correct equilibrium structure. Given the construction of the network that generates the initial positions, this seems quite likely, since it uses some canonical ordering (no permutation invariance) and does not include some notion of rotational symmetry. In contrast, there is previous work that demonstrates that this is possible with autoregressive models (Mansimov et al, Sci Rep, 2019; Gebauer et al, NeurIPS 2019). These methods also achieve much lower RMSDs than the presented approach on QM9 data (both <0.5 A RMSD). Another baseline would be to find a good positional guess, e.g. with RDkit, and then optimize with an ML force field that has already been used for MD (Schuett et al, J Chem Phys, 2019) or relaxation in crystal structure prediction (Podryabinkin, Phys. Rev. B, 2019).
Moreover, since the datasets without 3d positions are labeled with RDkit (which i guess uses the same method as in the RDKit baseline of Table 1?), it would be interesting to see whether these structures would also be sufficient for the property prediction of QM9 in Tab. 2. Currently, I am not convinced that the more accurate positions (w.r.t. DFT equilibrium) obtained by HamNet actually improve the prediction that much. In particular, since the improvement compared to prediction without conformations in Table 2 is not significant according to the reported error bars.

Pros
------
+ Interesting idea, clearly written paper
+ Physically motivated approach
+ Improvements against simple baselines and ablations

Cons
-------
- Missing of important baselines (see above)
- Only "multi-mae" reported for QM9. This makes it hard to judge the quality of the prediction and compare to predictions of individual properties in many other papers (DimeNet, MPNN, SchNet, HIP-NN, DTNN, PhysNet, etc). Please list also individual metrics and compare to previous work.
- Datasets without 3d positions are labeled with a computationally cheap method from RDKit.  This does not make much sense: Why then not directly use these positions?
- Timings for the prediction should be added to judge computational cost compared to a fast force-field or semi-empirical method.

Update:
The responses cleared up some aspects of the Hamiltonian engine and I adjusted my score accordingly. Still, additional baselines would improve the paper answer some open questions and validate some of the claims: Is the Ham. Eng. indeed faster than optimizing with a cheap force field? Is there then an advantage compared to featurization of the RDKit coordinates with one of the many MPNNs for 3d coordinates? This would be interesting to analyze.

---

> ### Author Response · Authors · 2020-11-22
> **Official reply to Reviewer 1**
>
> We thank you for the insightful comments. For the major comments, below are our responses:
>
> 1. **Why MD instead of structure optimization?**
>
> Firstly, as is discussed in Section 3.3, the MD-like dynamics is essentially an implicit process of structure  optimization: in fact, the Hamiltonian equations depict exactly the *gradient descending with momentum* (MGD) process. The process of MGD of optimizing $f(q)$ against $q$ follows
> $$
> q_t = q_{t-1} + \eta_t p_{t},
> $$
> $$
> p_t = p_{t-1} + \epsilon_t \nabla_q f(q).
> $$
> These descending steps are exactly the same as what Hamiltonian equations depict when $H(q; p)=T(p) + U(q)$ (let $f(q)=U(q)$) and $T(p)$ is the quadratic form of $p$ (so that $\partial H / \partial p \propto p$). Note that MGD is generally a more robust and faster optimization algorithm than simple GD.
>
> Secondly, different with current PES (potential energy surface) approaches , HamNet do not learn parameters in the potential functions from known (labeled) potential energies or forces. Instead, the output positions are directly supervised. This requires that the entire optimization process to be differentiable. Therefore, we implement the MD process, which is differentiable w.r.t the parameters in function $T$ and $U$. Also, another desired byproduct of using MD, however, is the better interpretability of the model.
>
> 2. **Why Hamiltonian Engine at all?**
>
> To address your concerns on 1) missing baselines of recent conformation generation algorithms (e.g. ETKDG, or CVGAE, G-SchNet et al as you mentioned); 2) (potentially) worse results than these baselines on conformation prediction tasks, we would like to emphasize that the ultimate goal of HamNet is to **derive good molecular fingerprints** rather than to *accurately predict the conformation in equilibrium*. Showing the results of Ham. Eng. vs RDKit on conformation prediction is to prove that the model correctly captures the conformation in $q,p \in R^{d_f}$, instead of arguing that Ham. Eng. is a state-of-the-art conformation prediction approach.
>
> **We believe to better understand the role of Ham. Eng., one may regard Ham. Eng. as a featurization approach from a encoder-decoder view**: the encoder calculates the hidden representations ($q^{(T)}, p^{(T)}$) from molecular graphs, and the decoder supervises $q^{(T)}$ with labeled conformations. The reason we need the Ham. Eng. is to better combine the structural and featural information (encoded in the molecular graph representations) with the spatial information (the labeled conformations) in the generalized space $R^{d_f}$.
>
> We also show by the experiment *HamNet (real conf.)* the benefits of using Ham. Eng.: even on QM9 where the real conformations are used, *HamNet (ours)* still outperforms *HamNet (real conf.)* (note that the dimensionality issue is addressed in the comparison, as real coordinations of atoms are expanded into $2 \times d_f$ so that two inputs have the same dimensionality). **That is, it is not the accuracy of conformations, but the featurization approach itself, that matters more.** This indicates the necessity of using Ham. Eng. to obtain the optimal results of fingerprints.
>
> Another potential benefits of using implicit $q,p$s compared with using 3D positions (say $x$s), is that the former enjoys better spatial invariance: i) the input of Ham. Eng. is spatial invariant (although the LSTM to some degree compromised the permutation invariance); ii) the output of Ham. Eng. is supervised with spatial invariant metrics (K-RMSD, ADJ-k). Therefore, as long as the initialization encoders (GCN+LSTM) are properly regularized, the Ham. Eng. learns a canonical space which enjoys spatial invariance.
>
> 3. **Why not using RDKit to generate conformations?**
>
> We do use RDKit conformations when the datasets lack real ones, but they are calculated *only for the training set*. Actually, in real scenarios, conformations of training sets are generally available; if not, at least the calculation is once-and-for-all. What HamNet saves is the time of conformation calculation in the *inference stage*, where labeled confs are NOT used. In the scenarios including drug candidate generation & large-scale virtual screening, this improvement is crucial.

---

### Author Response · Authors · 2021-03-01
**Fixed experimental results.**

We fixed some bugs in the experiments of Table 2, namely those of FreeSolv and ESOL.

As a result, some figures in Table 2 were altered, while the advantage of HamNet was still corroborated.

---

### Comment · ~Ziyao_Li1 · 2021-05-13
**Code available on Github.**

Our code is now available on Github with url:  https://github.com/PKUterran/HamNet

We are sorry that the code is somehow messy currently. Clean versions and a more detailed readme will be updated soon.

---

### Decision · Program_Chairs · 2021-01-07
**Final Decision**

**Decision:**

Accept (Poster)

**Comment:**

The paper proposes to lear molecular descriptors that account for the 3D structure of molecules. This is done by using first a "Hamiltonian Engine" that runs a brief simulation, predicting the structure of the small molecule by minimizing a learned potential energy, and second, a message passing algorithm that uses the predicted structure as input. The reported experimental results show state-of-the-art performance.

Strengths:

1 - Relevant contribution through the Hamiltonian Engine.

2 - Strong empirical results.

Weaknesses:

3 - Some reviewers mentioned that the readability of the paper could be improved.

I recommend the authors to also take into account the concerns of AnonReviewer1 to
improve the paper.